# Effects of Interaction between *Claroideogolmus etuicatum* and *Bacillus aryabhattai* on the Utilization of Organic Phosphorus in *Camellia oleifera* Abel.

**DOI:** 10.3390/jof9100977

**Published:** 2023-09-28

**Authors:** Yuxuan Huang, Yulan Lin, Linping Zhang, Fei Wu, Yang Zhang, Shaohua Huang

**Affiliations:** 1Key Laboratory of National Forestry and Grassland Administration for the Protectionand Restoration of Forest Ecosystem in Poyang Lake Basin, Jiangxi Agricultural University, Nanchang 330045, Chinahsh916@hotmail.com (S.H.); 2College of Meizhouwan Vocational Technology, Putian 351119, China; 3College of Life Sciences, Northwest Normal University, Lanzhou 730070, China

**Keywords:** arbuscular mycorrhizal fungi, phosphate solubilizing bacteria, oil tea, phosphorus mobilization

## Abstract

Arbuscular mycorrhizal fungi (AMF) and phosphate solubilizing bacteria (PSB) are involved in phosphorus (P) mobilization and turnover; however, the impact of their interaction on plant P absorption and organic P mineralization in the hyphosphere (rootless soil) are unknown. This study examined the interactive effects of two native microorganisms, namely *Claroideogolmus etuicatum* and *Bacillus aryabhattai*, and the effects of co-inoculation of both microorganisms on organic P mineralization and the subsequent transfer to *Camellia oleifera,* using a three-compartment microcosm with a nylon mesh barrier. The results demonstrated that the co-inoculation treatment (AMF + PSB) significantly increased the plant P content and biomass accumulation in *C*. *oleifera* compared to those of the non-inoculated control. Furthermore, co-inoculation boosted soil phosphatase and phytase activities as well as the liable P content. Compared to the non-inoculated control, inoculation of AMF decreased the NaOH-Po content. A correlation analysis showed that AMF colonization and hyphal density was significantly positively correlated with H_2_O-P and NaHCO_3_-Pi and negatively correlated with NaOH-Po. It was shown that co-inoculation could increase phosphatase activity, phytase activity, and promote the liable P content, thus increasing the phosphorus content and biomass accumulation of *C*. *oleifera*. In conclusion, AMF and PSB interactively enhanced the mineralization of soil organic P, and therefore positively affected P uptake and plant growth.

## 1. Introduction

Phosphorus (P) is a crucial component of ecosystems, and it plays an important role in promoting agroforestry productivity [1,2]. Two-thirds of the world’s soils are phosphorus deficient, which limits the growth of crops; the application of phosphorus fertilizer is an essential way to boost the productivity of agroforestry crops [3,4]. However, there are many issues with phosphorus fertilizer application in agroforestry practices. On the one hand, about 80% of the world’s phosphorus rock resources are used for the production of phosphorus fertilizer, and phosphorus rock is a non-renewable resource [5]. On the other hand, the utilization efficiency of soil phosphorus is extremely low, and about 70% of phosphorus fertilizers are easily and rapidly combined by cations such as Ca^2+^, Mg^2+^, Al^3+^, Fe^3+^, and transformed into insoluble phosphorus that is difficult for plants to utilize [6,7]. Therefore, understanding how to release the ineffective state of phosphorus stock in soil through biological or ecological pathways is of great significance to improve soil phosphorus effectiveness and to maintain ecosystem stability.

Soil microorganisms are key drivers of soil functional processes, including organic matter decomposition, nutrient turnover, and nutrient release, especially of nitrogen (N) or phosphorus (P), for subsequent plant capture [8]. Arbuscular mycorrhizal fungi (AMF) are a class of soil microorganisms which can form a symbiosis with the roots of 80% of land plants [9]. AMF can deliver mineral nutrients, especially phosphorus, to host plants; therefore, they have garnered attention as potential biofertilizers [10]. AMF are capable of releasing protons to migrate insoluble soil phosphate and extend their extensive hyphae from phosphorus-depleted zones in order to explore a larger volume of soil for inorganic phosphorus sources [11]. Similar to AMF, phosphate solubilizing bacteria (PSB) are a group of microorganisms that can convert insoluble phosphorus compounds into active forms by releasing organic acids. In addition, PSB can produce hormones such as cytokinin and indoleacetic acid that promote plant growth. [12]. Efficient PSB has the potential to replace chemical fertilizers to reduce environmental pollution and to promote ecological balance [13]. Previous studies have shown that co-inoculation of AMF and PSB can increase crop yields [14,15], the number of PSB [16], and phosphatase activity [17,18]. However, most of these results have not necessarily reflected the interaction between AMF and PSB, because plant roots cannot be ruled out. The effects of PSB inoculation between mycelia on plant phosphorus uptake and soil phosphorus fractions are unclear, and understanding the direct interactions between AMF and PSB may help to regulate specific microorganisms to enhance soil phosphorus effectiveness [19,20].

*Camellia oleifera* Abel. (Theaceae), one of the world’s four famous woody oil plants, is a unique edible economic tree species in China [21]. *C*. *oleifera* has an unsaturated fatty acid content of up to 90%, which is much higher than that of vegetable oil, peanut oil, and soybean oil, and which has high economic and nutritional value [22]. At present, China’s vegetable oil self-sufficiency rate has been declining year by year, and the import of vegetable oil is among the forefront in the world. Promoting the development of the *C*. *oleifera* industry can alleviate the contradiction between the supply and demand of vegetable oil in China. However, the effective phosphorus content of acidic soil in southern China is very low, which limits the productivity of *C*. *oleifera* [6]. Previous studies have shown that inoculation of AMF (*Claroideogolmus etuicatum*) promoted the mineralization of organic phosphorus in oil tea soil and positively affected root phosphatase activity [23]. In addition, the PSB strain JX285 (*Bacillus aryabhattai*) isolated from the inter-root of *C*. *oleifera* has been shown to promote *C*. *oleifera* growth and nutrient uptake [6,24]. The current studies have been conducted using only single inoculation treatments (AMF or PSB). However, is there any competition or mutualism between AMF and PSB? What are the effects on *C*. *oleifera* phosphorus uptake and soil phosphorus conversion? In this study, we investigated the effects of the interaction between AMF and PSB on P uptake, soil phosphatase activity, and soil phosphorus fractions by inoculating PSB into the hyphosphere, using a three-compartment culture. The results may provide a theoretical basis for the development of microbial fertilizers for use in agroforestry systems.

## 2. Materials and Methods

### 2.1. Biological Materials and Growth Substrates

The *C*. *oleifera* seeds were provided by the Jiangxi Academy of Forestry Research, China. First, the seeds were sterilized and cleaned with potassium permanganate and sterile water, then they were germinated on wet gauze at 30 °C, transplanted to pots with sterilized sand after seed development, and finally uniform seedlings were selected and removed to the three-compartment device.

The AMF strain (*Claroideogolmus etuicatum*) was provided by Professor Qiangsheng Wu, Institute of Root Biology, Yangtze University, Hubei. The PSB strain used was *Bacillus aryabhattai* (JX285), which was provided by the Forest Pathology Group, Jiangxi Agricultural University (Nanchang, China).

The growth medium used in this experiment was composed of soil and sand (1:1 *v*/*v*). The soil-sand mixture was autoclaved at 121 °C for 4 h. The growth medium contained 41.25 g organic matter kg^−1^, 2.50 mg effective phosphorus·kg^−1^, 21.2 mg fast-acting potassium kg^−1^, 14.24 mg ammonium nitrogen (NH_4_^+^) kg^−1^, and 2.26 mg nitrate nitrogen (NO_3_^−^) kg^−1^.

### 2.2. Three-Compartment Device and Experimental Design

A polyvinyl chloride (PVC) plexiglass block was used to make a three-compartment culture system to meet the experimental requirements (Figure 1). The culture system consisted of a root compartment (10 ×15 × 14 cm), buffer zone (2 × 15 × 14 cm), and hyphal compartment (5 × 15 × 14 cm). The root and hyphal compartments were separated from each other by a 30 mm nylon mesh. 

A pot experiment was conducted using a two-factor completely randomized block design consisting of four inoculation treatments and two P treatments. The inoculation treatments were as follows: inoculated with *C. etuicatum* (CE), inoculated with *Bacillus aryabhattai* (BA), inoculated with the mixture of *C. etuicatum* and *Bacillus aryabhattai* (CE + BA), inoculated with autoclaved inoculum (CK). The P treatments were as follows: no calcium phytate (C_6_H_16_Ca_24_O_24_P_6_) (0 mg·kg^−1^) and the addition of calcium phytate (C_6_H_16_Ca_24_O_24_P_6_) (75 mg·kg^−1^). Each treatment included 15 replicates. 

The seedings were planted in root compartments filled with 1.5 kg of growth medium. Each root compartment was inoculated with a 70 g mixture of sand and *C. etuicatum* (30 spores per g) (*n* = 60), and control treatments added 70 g of autoclaved inoculum. The buffer zone was filled with 800 g of sterilized sand. The hyphal compartment was filled with 1.5 kg of sterilized sand and 0.1125 g of calcium phytate treatment (*n* = 60). The hyphal compartment was inoculated with 10 mL of mixed bacterial suspensions, and control treatments added an equal volume of inactivated bacterial liquid. The pots were placed in a greenhouse with 12 h of light per day, at Jiangxi Agricultural University, from May 2019 to May 2020.

### 2.3. Mycorrhizal Colonization and Hyphal Density

Mycorrhizal colonization was determined by Phillips and Hayman [25]. Hyphal density was measured according to Abbott et al. [26].

### 2.4. Plant Growth and P Content Measurement

Plant height was measured with a measuring tape (Sata, Shanghai, China). The leaves, stems, and roots of *C*. *oleifera* were harvested separately, dried to a constant weight at 75 °C, and weighed to calculate the total biomass of each plant.

The dried stems and roots were thoroughly ground and homogenized, and the P content was determined by molybdenum blue spectrophotometry.

### 2.5. Measurement of Substrate Phosphatase Activities and P Content

The activities of soil acid phosphatase (S-ACP), alkaline phosphatase (S-ALP), and phytase activity (S-phytase) were measured by corresponding assay kits (Suzhou Keming Biological Co., Ltd., Suzhou, China), separately.

The contents of the different forms of phosphorus including H_2_O-P, NaHCO_3_^−^Pi, and NaHCO_3_^−^Po were determined according to the method by Hedley et al. [27].

### 2.6. Data Analysis

The statistical analyses were performed using the SPSS software, version 20.0 (SPSS Inc., Chicago, IL, USA). The Kolmogorov–Smirnov test and Levene’s test were used to check the normality and chi-square of the data, respectively. A two-way ANOVA was performed to test the significance of inoculation application, organic P application, and their interaction. A one-way ANOVA was used to test the differences among the different inoculation treatments under the same P treatment. Means were compared with Duncan’s multiple range test at the 5% level.

## 3. Results

### 3.1. Mycorrhizal Colonization and Hyphal Density

The results showed that the seedlings without inoculation of *C. etuicatum* did not form any mycorrhizae. In the absence of organophosphorus (P0, 0 mg·kg^−1^), the AMF colonization rate of co-inoculation was significantly higher than that of single inoculation (*p* < 0.05) (Figure 2). However, the AMF colonization of co-inoculation was significantly lower than that of the single inoculation of AMF after the addition of organophosphorus. The highest colonization rate (50.67 ± 3.21%) was found in the treatment with the single inoculation of *C. etuicatum* at the P75 (75 mg·kg^−1^) level. Interestingly, the lowest colonization rate (36.67 ± 2.08%) was also for the single inoculation CE treatment at the P0 level. Thus, PSB inoculation inhibited mycorrhizal colonization under phosphorus-deprived conditions and promoted mycorrhizal colonization under phosphorus-sufficient conditions.

Under the two phosphorus levels (0 mg·kg^−1^ and 75 mg·kg^−1^), the hyphal density of the single inoculation of AMF was significantly (*p* < 0.05) higher than that of the co-inoculation treatment, and inoculation of PSB significantly reduced hyphal density by 31.67% and 25.25%, respectively. 

### 3.2. Plant Height and Biomass

The organic P treatment and inoculation treatments significantly (*p* < 0.05) influenced plant height and dry weight (Table 1). At the two P levels, the plant height of *C*. *oleifera* did not significantly (*p* > 0.05) differ under the single inoculation of AMF or PSB; moreover, plant height was significantly (*p* < 0.05) higher under the co-inoculation treatment than the single treatment. Inoculation of AMF and the co-inoculation treatment significantly (*p* < 0.05) increased the dry weight of *C*. *oleifera*, while inoculation of PSB increased the shoot dry weight at the P0 level.

### 3.3. P Content of C. oleifera

The two-way ANOVA results showed that the inoculation treatments significantly (*p* < 0.05) influenced the P content of *C*. *oleifera* (Table 2). Compared to the non-inoculated control, the inoculation treatments with AMF or PSB, and the co-inoculation treatment all significantly (*p* < 0.05) increased the shoot and root P content of *C*. *oleifera*, and the effect of the co-inoculation treatment was significantly (*p* < 0.05) higher than those of the single inoculation of AMF or PSB under two organic P treatment (Figure 3). At the P0 level, compared to the single inoculation of AMF, the single inoculation of PSB significantly (*p* < 0.05) increased the P content. However, compared to the inoculation of PSB, the inoculation of AMF significantly (*p* < 0.05) increased root P content under the P75 level. Interestingly, P application significantly (*p* < 0.05) increased root P content but had no significant effect on shoot P content under all inoculation treatments. 

### 3.4. Soil Phosphatase and Phytase Activities

The inoculation treatments significantly (*p* < 0.05) influenced the soil phosphatase activities (Table 3). At the P0 level, compared to the non-inoculated control, the inoculation treatments of AMF or PSB and the co-inoculation treatment all significantly (*p* < 0.05) increased the S-ACP activity by 2.92%, 6.08%, and 6.59%; the S-ALP activity by 8.47%, 9.72%, and 6.62%; and the S-phytase activity by 22.78%, 45.50%, and 53.80%, respectively (Figure 4). At the P75 level, compared to the non-inoculated control, the inoculation treatments with AMF or PSB and the co-inoculation treatment all significantly increased S-ACP activity and S-phytase activity. Compared to others inoculation treatments, the co-inoculation treatment significantly (*p* < 0.05) increased S-ALP activity. 

### 3.5. Soil Liable P Content

The organic P treatment and inoculation treatment significantly (*p* < 0.05) influenced the soil liable P content (Table 3). The co-inoculation treatment had a higher liable P content than the single and non-inoculated treatments under the two P levels (Figure 5), and compared to the non-inoculated control, the co-inoculation treatment increased H_2_0-P content by 39.31% and 48.89%, respectively. Interestingly, inoculation of PSB was more effective in significantly (*p* < 0.05) increasing NaHCO_3_-Po content than the inoculation of AMF under the two P level. 

### 3.6. Soil Moderately Liable P Content

The organic P treatment and inoculation treatments significantly (*p* < 0.05) influenced the soil moderately liable P content (Table 3). The co-inoculation treatment had a higher NaOH-Pi content than the single and non-inoculated treatments under the two P level (Figure 6A). Interestingly, the inoculation treatments with AMF or PSB and the co-inoculation treatment all significantly (*p* < 0.05) increased NaOH-Pi content. Inoculation treatment with PSB had a higher NaOH-Po content than other inoculation treatments under each P level (Figure 6B). However, the co-inoculation treatment had a lower NaOH-Po content than the non-inoculated treatment under the P0 level. Compared to the non-inoculated control, inoculation of AMF decreased the NaOH-Po content under the two P levels.

### 3.7. Correlation Analysis

The correlation analysis showed that soil phosphatase and phytase activities were positively correlated with plant phosphorus content (Figure 7). H_2_O-P and NaHCO_3_-Pi were significantly (*p* < 0.05) and positively correlated with AMF colonization, hyphal density, plant growth, and P content. NaOH-Po was significantly (*p* < 0.05) negatively correlated with AMF colonization and hyphal density. The root and shoot P content were significantly (*p* < 0.05) and positively correlated with S-ACP, S-ALP, S-phytase, H_2_O-P, NaHCO_3_-Pi, NaHCO_3_-Po, and NaOH-Pi. AMF colonization was significantly (*p* < 0.05) and positively correlated with S-ACP, S-ALP, S-phytase, H_2_O-*p*, NaHCO_3_-Pi, and NaOH-Pi.

## 4. Discussion

### 4.1. PSB Addition Limits AMF Mycelial Growth and C. oleifera Colonization

Root colonization of AMF can be enhanced in the presence of some rhizobacteria, which promote plant growth and nutrient access [28]. Fester et al. [29] showed that the addition of three potential mycorrhiza-helper bacteria (MHB) increased mycorrhizal root colonization by two–three times. Saxena et al. [30] showed that *Burkholderia cepacia* significantly increased AMF (*Glomus etunicatum*) root colonization. However, in this study, the addition of PSB significantly reduced the hyphal density of AMF in the hyphal compartment, and the addition of PSB reduced AMF root colonization under the P75 level. The decreased AMF hyphal growth and root colonization due to PSB addition was probably because the bacteria were not mycorrhiza-helper bacteria like *Pseudomonas fluorescens*, which can secrete cell wall degrading enzymes and soften the root cell wall, thus making it easier for AMF to penetrate the roots [31,32]. In this study, there may have been some competition between PSB and AMF, because AMF shared the carbon source fixed by plant photosynthesis with PSB, which inhibited the growth of their own mycelium.

### 4.2. AMF and PSB Interactions Promote C. oleifera Growth and Phosphorus Uptake

Both AMF and PSB have positive effects on phosphorus uptake and growth of *C*. *oleifera*. Wu et al. [23] showed that inoculation of AMF could improve root activity, as well as enhance the acquisition of organic phosphorus and the accumulation of biomass in *C*. *oleifera*. Wu et al. [6] showed that inoculation of PSB increased the N and P contents of the leaves and promoted the growth of *C*. *oleifera*. AMF and PSB may interact with each other in soil phosphorus utilization. PSB can attach to the root system and extraradical hyphae of AMF [33], which promotes plants’ access to nutrients and improves crop yield [30]. In this study, co-inoculation of AMF and PSB significantly increased the P content of *C*. *oleifera* and promoted growth and biomass accumulation compared with the non-inoculated control and single inoculation treatments. Liu et al. [15] showed that co-inoculation of AMF and PSB could promote growth and P uptake of *Medicago sativa* L. On the one hand, AMF increases the absorbing contact area of plant roots by forming an extensive hyphal network in the soil [34,35]. On the other hand, PSB converts insoluble P in the soil into usable P that can be absorbed and utilized by plants, which improves plant P nutrition and promotes plant growth [36,37]. These points also apply in this experiment. Previous studies have mainly focused on the positive effects of PSB on P content in the rhizosphere, and this study further confirmed that, when PSB is far away from the rhizosphere, PSB can also play its ecological role through the extra-root hyphae of AMF.

### 4.3. AMF Interacts with PSB to Enhance Soil Phosphatase Activity for Organic P Mineralization

Although soils contain large amounts of P, the majority of this P is unavailable to plants [38]. The majority of soil organic P is immobilized and adsorbed by some metal ions or minerals, forming stable compounds that must be hydrolyzed with phosphatases to release more soluble inorganic P for plant uptake and utilization [36,38]. The results of this study showed that co-inoculation increased soil phosphatase activities, phytase activity, and liable P content in the hyphal compartment. There is no evidence that AMF can directly participate in the hydrolysis of organic P, and it is controversial whether AMF can secrete phosphatase [39]. However, the addition of AMF can enhance soil phosphatase activities and increase soil effective P content, which has been proven in some agroforestry crops [15,40]. Wang et al. [40] found that hyphosphere acidification induced by the AMF mycelium resulted in enhanced phosphatase activities and mineralization of phytin. AMF can increase the concentration of carbon in the hyphosphere by releasing secretion, which attracts the colonization of PSB [33,41]. PSB achieves the effect of P solubilization through the production of organic acids, phosphatases, and H^+^ that not only reduce soil pH, but also chelate with metal ions such as Ca, Al, and Fe, converting insoluble P into effective phosphorus that is easily absorbed by plants for their utilization [42]. Soil NaOH-Po belongs to the moderately active state of P which is basically ineffective for plants [43], and can be converted into effective P available for plant uptake through various biological and physical chemical reactions in soil. This study showed that co-inoculation increased S-ACP, S-ALP, and S-phytase activities, decreased soil NaOH-Po content, and increased liable P content. In addition, the correlation analysis showed that AMF colonization and hyphal density was significantly positively correlated with H_2_O-P and NaHCO_3_-Pi, and negatively correlated with NaOH-Po, which suggests that *C*. *etuicatum* and *B*. *aryabhattai* co-inoculation may convert soil NaOH-Po into plant-absorbable liable P to promote plant P acquisition and growth.

## 5. Conclusions

In this study, we investigated the effects of single and co-inoculation of AMF (*C. etuicatum*) and PSB (*B. aryabhattai*) in the hyphal compartments on plant growth and soil phosphorus fractions. The results showed that co-inoculation had positive effects on plant growth, plant phosphorus content, soil phosphatase activities, and soil phytase activity. Co-inoculation of AMF and PSB decreased soil NaOH-Po content and increased liable P content, thereby promoting plant P uptake. The use of AMF and PSB as inoculants may provide an alternative to chemical fertilizers, and therefore promote sustainable agroforestry.

## Figures and Tables

**Figure 1 jof-09-00977-f001:**
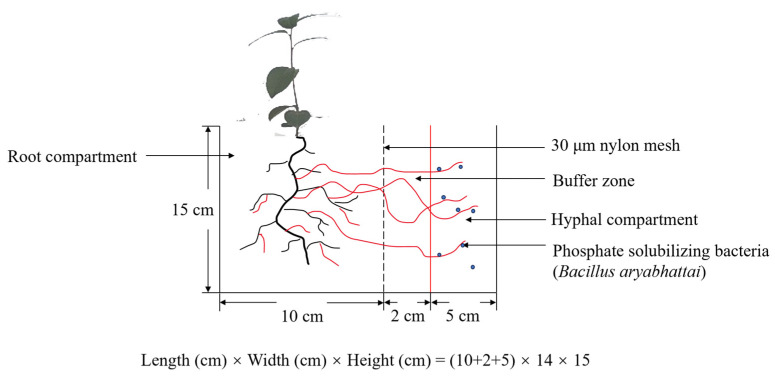
The three-compartment culture system.

**Figure 2 jof-09-00977-f002:**
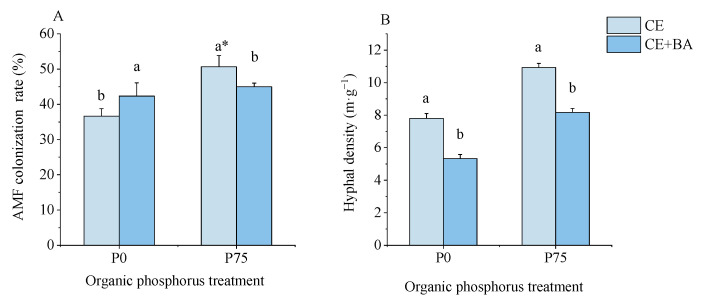
Effects of the organic P treatment and inoculation treatment on the AMF colonization and hyphal density of *C. oleifera.* Note: (**A**) shows the effect of organophosphorus treatment and inoculation treatment on AMF colonization. (**B**) shows the effect of organophosphorus treatment and inoculation treatment on Hyphal denisty. Data are means ± SD (*n* = 3). CE, inoculated with *C. etuicatum*; CE + BA, inoculated with *C. etuicatum* and *B. aryabhattai*; P0, no organic phosphate fertilizer; P75, added 75 mg·kg^−1^ organic phosphate fertilizer. Different letters indicate significant differences between different inoculation treatments under the same phosphorus level at *p* < 0.05; asterisks indicate significant differences in phosphorus levels under the same inoculation treatment at *p* < 0.05.

**Figure 3 jof-09-00977-f003:**
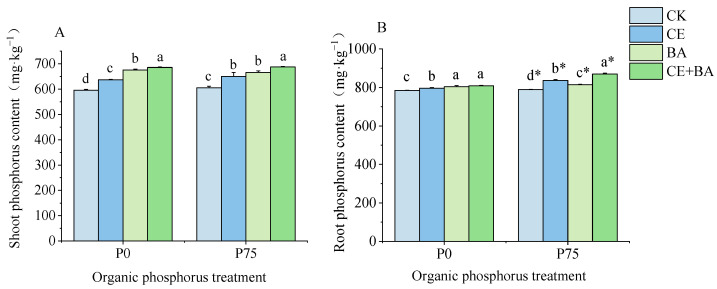
Effects of organic P treatment and inoculation treatment on the plant P content of *C. oleifera*. Note: (**A**) shows the effects of organic P treatment and inoculation treatment on the shoot P content of *C. oleifera*. (**B**) shows the effects of organic P treatment and inoculation treatment on the shoot P content of *C. oleifera*. Data are means ± SD (*n* = 3). CK, non-inoculated control; CE, inoculated with *C. etuicatum*; BA, inoculated with *B. aryabhattai*; CE + BA, inoculated with *C. etuicatum* and *B. aryabhattai*; P0, no organic phosphate fertilizer; P75, added 75 mg·kg^−1^ organic phosphate fertilizer. Different letters indicate significant differences between different inoculation treatments under the same phosphorus level at *p* < 0.05; asterisks indicate significant differences in phosphorus levels under the same inoculation treatment at *p* < 0.05. *, Significance level *p* < 0.05.

**Figure 4 jof-09-00977-f004:**
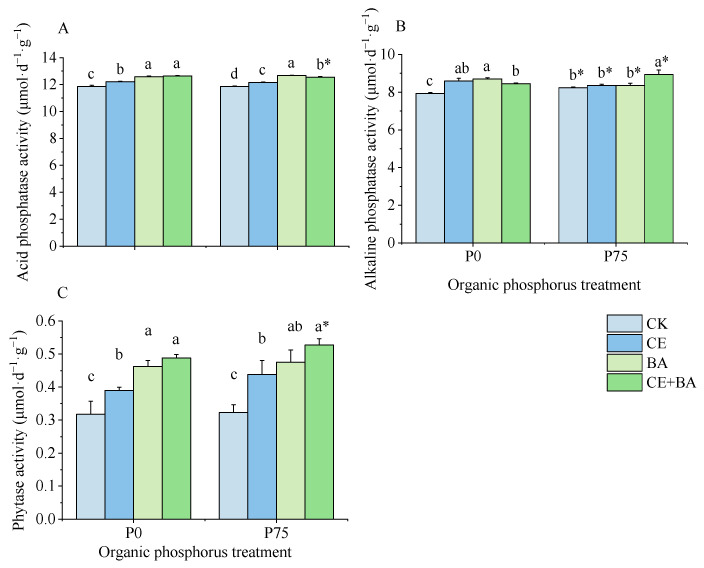
Effects of organic P treatment and inoculation treatment on the soil phosphatase activities of *C. oleifera*. Note: (**A**) shows the effects of organic P treatment and inoculation treatment on the soil acid phosphatase (S-ACP) activities of *C. oleifera*. (**B**) shows the effects of organic P treatment and inoculation treatment on the soil alkaline phosphatase (S-ALP) activities of *C. oleifera*. (**C**) shows the effects of organic P treatment and inoculation treatment on the soil phytase activity(S-phytase) activities of *C. oleifera*. Data are means ± SD (*n* = 3). CK, non-inoculated control; CE, inoculated with *C. etuicatum*; BA, inoculated with *B. aryabhattai*; CE + BA, inoculated with *C. etuicatum* and *B. aryabhattai*; P0, no organic phosphate fertilizer; P75, added 75 mg·kg^−1^ organic phosphate fertilizer. Different letters indicate significant differences between different inoculation treatments under the same phosphorus level at *p* < 0.05; asterisks indicate significant differences in phosphorus levels under the same inoculation treatment at *p* < 0.05.

**Figure 5 jof-09-00977-f005:**
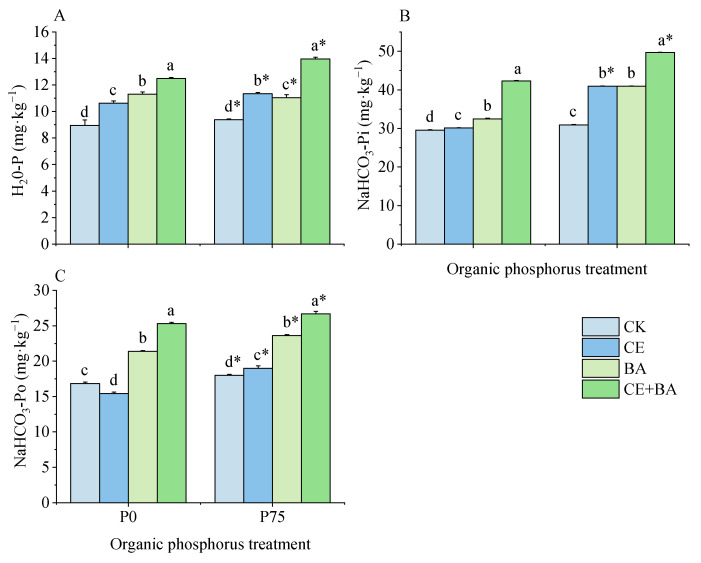
Effects of organic P treatment and inoculation treatment on the soil liable P content of *C. oleifera*. Note: (**A**) shows the effects of organic P treatment and inoculation treatment on the content of H_2_O-P of *C. oleifera*. (**B**) shows the effects of organic P treatment and inoculation treatment on the content of NaHCO_3_-Pi of *C. oleifera*. (**C**) shows the effects of organic P treatment and inoculation treatment on the content of NaHCO_3_-Po of *C. oleifera*. Data are means ± SD (*n* = 3). CK: non-inoculation control; CE: inoculated with *C. etuicatum*; BA: inoculated with *B. aryabhattai*; CE + BA, inoculated with *C. etuicatum* and *B. aryabhattai*; P0, no organic phosphate fertilizer; P75, added 75 mg·kg^−1^ organic phosphate fertilizer. Different letters indicate significant differences between different inoculation treatments under the same phosphorus level at *p* < 0.05; asterisks indicate significant differences in phosphorus levels under the same inoculation treatment at *p* < 0.05.

**Figure 6 jof-09-00977-f006:**
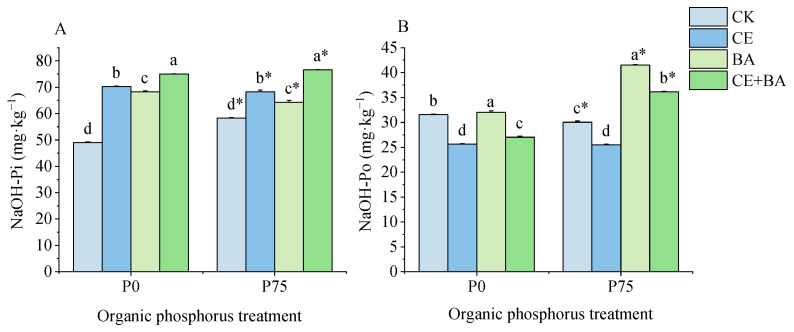
Effects of organic P treatment and inoculation treatment on the soil moderately liable P content of *C. oleifera*. Note: (**A**) shows the effects of organic P treatment and inoculation treatment on the content of NaOH-Pi of *C. oleifera*. (**B**) shows the effects of organic P treatment and inoculation treatment on the content of NaOH-Po of *C. oleifera*. Data are means ± SD (*n* = 3). CK, non-inoculated control; CE, inoculated with *C. etuicatum*; BA, inoculated with *B. aryabhattai*; CE + BA, inoculated with *C. etuicatum* and *B. aryabhattai*; P0, no organic phosphate fertilizer; P75, added 75 mg·kg^−1^ organic phosphate fertilizer. Different letters indicate significant differences between different inoculation treatments under the same phosphorus level at *p* < 0.05; asterisks indicate significant differences in phosphorus levels under the same inoculation treatment at *p* < 0.05.

**Figure 7 jof-09-00977-f007:**
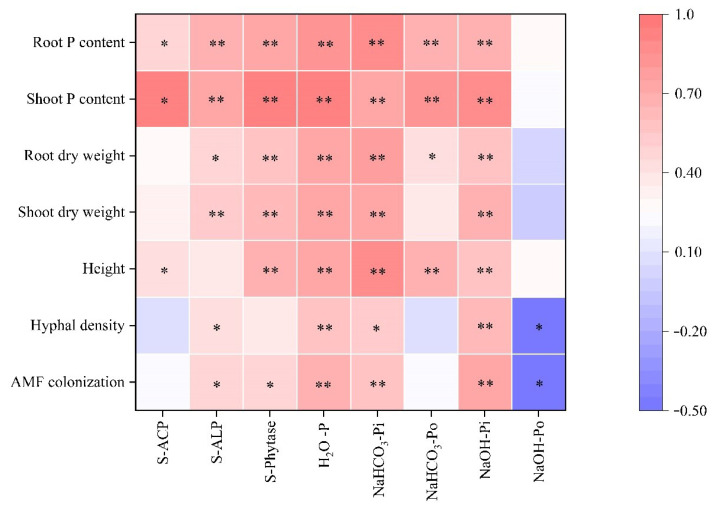
Correlation analysis of soil phosphatase activities, P content, and plant physiological indexes. Note: Data are Pearson correlation coefficients (*n* = 24). **, significant effect at *p* < 0.01; *, significant effect at *p* < 0.05. blank space, no significant effect.

**Table 1 jof-09-00977-t001:** Effects of organic P treatment and inoculation treatment on the plant height and dry weight of *C. oleifera*.

P	Inoculation	Plant Height cm	Shoot Dry Weight g	Root Dry Weight g
P0	CK	12.63 ± 0.23 b	0.74 ± 0.04 b	0.63 ± 0.10 b
	CE	13.03 ± 0.84 b	0.95 ± 0.03 a	1.20 ± 0.22 a
	BA	13.53 ± 0.47 b	0.94 ± 0.09 a	0.74 ± 0.08 b
	CE + BA	15.30 ± 0.20 a	1.06 ± 0.12 a	1.08 ± 0.18 a
P75	CK	14.30 ± 1.15 b	0.86 ± 0.16 b	0.46 ± 0.02 c *
	CE	16.40 ± 0.40 a *	1.86 ± 0.17 a *	1.74 ± 0.17 a *
	BA	15.60 ± 0.70 a *	1.05 ± 0.25 b	1.05 ± 0.13 b *
	CE + BA	16.93 ± 0.21 a *	1.93 ± 0.09 a *	1.79 ± 0.11 a *
Two-way ANOVA				
P		75.185 **	81.637 **	36.747 **
Inoculation		18.747 **	35.750 **	60.061 **
P × Inoculation		2.607 NS	16.655 **	11.227 **

Note: Data are means ± SD (*n* = 3). CK, non-inoculated control; CE, inoculated with *C. etuicatum*; BA, inoculated with *B. aryabhattai*; CE + BA, inoculated with *C. etuicatum* and *B. aryabhattai*; P0, no organic phosphate fertilizer; P75, added 75 mg·kg^−1^ organic phosphate fertilizer. Different letters indicate significant differences between different inoculation treatments under the same phosphorus level at *p* < 0.05; asterisks indicate significant differences in phosphorus levels under the same inoculation treatment at *p* < 0.05. * Significance level *p* < 0.05, ** significance level, and NS, no significant effect.

**Table 2 jof-09-00977-t002:** Results of two-way ANOVA for the effects of organic P treatment (P), inoculation treatment (Inoculation), and their interaction on P content and phosphatase activities.

Index	P	Inoculation	P × Inoculation
Shoot P content	2.217 NS	191.010 **	3.595 *
Root P content	463.154 **	253.411 **	91.575 **

Note: Data are equality of variances. *, Significance level *p* < 0.05; **, significance level; NS, no significant effect.

**Table 3 jof-09-00977-t003:** Results of two-way ANOVA for the effects of organic P treatment (P), inoculation treatment (Inoculation), and their interaction on phosphatase activities and phosphorus content.

Index	P	Inoculation	P × Inoculation
S-ACP activity	0.211 NS	337.277 **	4.153 *
S-ALP activity	1.237 NS	32.167 **	20.049 **
S-phytase activity	5.559 *	51.282 **	0.802 NS
H_2_0-P	53.255 **	424.131 **	19.824 **
NaHCO_3_-Pi	16,946.042 **	14,928.094 **	1406.401 **
NaHCO_3_-Po	508.959 **	2092.466 **	34.980 **
NaOH-Pi	54.846 **	3193.198 **	316.006 **
NaOH-Po	3378.228 **	3975.991 **	1615.785 **

Note: *, Significance level *p* < 0.05; **, significance level; NS, no significant effect.

## Data Availability

Not applicable.

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
