# Peer review of "Effects of Interaction between *Claroideogolmus etuicatum* and *Bacillus aryabhattai* on the Utilization of Organic Phosphorus in *Camellia oleifera* Abel."

_jof, 2023, doi:10.3390/jof9100977_

Round 1

Reviewer 1 Report

It will be helpful if you clearly describe the materials and methods in detail especially, regarding the various treatment combinations and the total number of experimental units.

There are lots of long and winding sentences whose meanings get lost in context. There is the need to rewrite these sentences as short as possible to retain or clearly capture the intended meanings. I have shared a couple of these sentences in the attached review comments.

Author Response

Thank you

Reviewer 2 Report

The manuscript by Huang et al. is devoted to the study of the influence of C. etuicatum and B. aryabhattai co-inoculation on organic P mineralization and transfer to Camellia oleifera. This work contains very interesting results. The text is well written; Materials and Methods are described in detail; the Introduction and Discussion sections are brief but sufficient.

However, I have a few comments:

1. It seems that some information is written twice in the Abstract section. To avoid this, it is necessary to change its structure.

2. Paragraph 2.3. – it is necessary to briefly describe it.

3. There are typos in the text. For example, Table 1, line 181, Table 2, Table 3:Two-way ANONVA”.

Please check the text for incorrect words and phrases.

4. Please improve the quality of the figures (especially Figure 1). Don't use screenshots.

Author Response

thank you

Reviewer 3 Report

The reviewed research is interesting and has an important aspect, not only meritorical but also practical. The following comments are intended to highlight the importance of the research.

Lines 72-78: This fragment is very important for the rest of the text, so it should be better highlighted. It would be much better to pose hypotheses rather than questions.

Line 198: Please explain the abbreviations: CK, CE, BA, CE+BA in the caption under Figure 3. Same line 214: Figure 4, line 225: Figure 5, line 237: Figure 6

Line 310: Conclusions This part of the text is not sufficient in relation to the purpose of the research. The results are interesting, so I would expect a broader discussion in the conclusions.

Author Response

thanks
